# Fluctuation Theorem of Information Exchange within an Ensemble of Paths Conditioned on Correlated-Microstates

**DOI:** 10.3390/e21050477

**Published:** 2019-05-07

**Authors:** Lee Jinwoo

**Affiliations:** Department of Mathematics, Kwangwoon University, 20 Kwangwoon-ro, Seoul 01897, Korea; jinwoolee@kw.ac.kr

**Keywords:** local non-equilibrium thermodynamics, fluctuation theorem, mutual information, entropy production, local mutual information, thermodynamics of information, stochastic thermodynamics

## Abstract

Fluctuation theorems are a class of equalities that express universal properties of the probability distribution of a fluctuating path functional such as heat, work or entropy production over an ensemble of trajectories during a non-equilibrium process with a well-defined initial distribution. Jinwoo and Tanaka (Jinwoo, L.; Tanaka, H. *Sci. Rep.*
**2015**, *5*, 7832) have shown that work fluctuation theorems hold even within an ensemble of paths to each state, making it clear that entropy and free energy of each microstate encode heat and work, respectively, within the conditioned set. Here we show that information that is characterized by the point-wise mutual information for each correlated state between two subsystems in a heat bath encodes the entropy production of the subsystems and heat bath during a coupling process. To this end, we extend the fluctuation theorem of information exchange (Sagawa, T.; Ueda, M. *Phys. Rev. Lett.*
**2012**, *109*, 180602) by showing that the fluctuation theorem holds even within an ensemble of paths that reach a correlated state during dynamic co-evolution of two subsystems.

## 1. Introduction

Thermal fluctuations play an important role in the functioning of molecular machines: fluctuations mediate the exchange of energy between molecules and the environment, enabling molecules to overcome free energy barriers and to stabilize in low free energy regions. They make positions and velocities random variables, and thus make path functionals such as heat and work fluctuating quantities. In the past two decades, a class of relations called fluctuation theorems have shown that there are universal laws that regulate fluctuating quantities during a process that drives a system far from equilibrium. The Jarzynski equality, for example, links work to the change of equilibrium free energy [1], and the Crooks fluctuation theorem relates the probability of work to the dissipation of work [2] if we mention a few. There are many variations on these basic relations. Seifert has extended the second-law to the level of individual trajectories [3], and Hatano and Sasa have considered transitions between steady states [4]. Experiments on single molecular levels have verified the fluctuation theorems, providing critical insights on the behavior of bio-molecules [5,6,7,8,9,10,11,12,13].

Information is an essential subtopic of fluctuation theorems [14,15,16]. Beginning with pioneering studies on feedback controlled systems [17,18], unifying formulations of information thermodynamics have been established [19,20,21,22,23]. Especially, Sagawa and Ueda have introduced information to the realm of fluctuation theorems [24]. They have established a fluctuation theorem of information exchange, unifying non-equilibrium processes of measurement and feedback control [25]. They have considered a situation where a system, say *X*, evolves in such a manner that depends on state *y* of another system *Y* the state of which is fixed during the evolution of the state of *X*. In this setup, they have shown that establishing a correlation between the two subsystems accompanies an entropy production. Very recently, we have released the constraint that Sagawa and Ueda have assumed, and proved that the same form of the fluctuation theorem of information exchange holds even when both subsystems *X* and *Y* co-evolve in time [26].

In the context of fluctuation theorems, external control $\lambda_{t}$ defines a process by varying the parameter in a predetermined manner during 0≤t≤τ. One repeats the process according to initial probability distribution $P_{0}$, and then, a system generates as a response an ensemble of microscopic trajectories $\left\{x_{t}\right\}$. Jinwoo and Tanaka [27,28] have shown that the Jarzynski equality and the Crooks fluctuation theorem hold even within an ensemble of trajectories conditioned on a fixed microstate at final time τ, where the local form of non-equilibrium free energy replaces the role of equilibrium free energy in the equations, making it clear that free energy of microstate $x_{\tau }$ encodes the amount of supplied work for reaching $x_{\tau }$ during processes $\lambda_{t}$. Here local means that a term is related to microstate *x* at time τ considered as an ensemble.

In this paper, we apply this conceptual framework of considering a single microstate as an ensemble of trajectories to the fluctuation theorem of information exchange (see Figure 1a). We show that mutual information of a correlated-microstates encodes the amount of entropy production within the ensemble of paths that reach the correlated-states. This local version of the fluctuation theorem of information exchange provides much more detailed information for each correlated-microstates compared to the results in [25,26]. In the existing approaches that consider the ensemble of all paths, each point-wise mutual information does not provide specific details on a correlated-microstates, but in this new approach of focusing on a subset of the ensemble, local mutual information provides detailed knowledge on particular correlated-states.

We organize the paper as follows: In Section 2, we briefly review some fluctuation theorems that we have mentioned. In Section 3, we prove the main theorem and its corollary. In Section 4, we provide illustrative examples, and in Section 5, we discuss the implication of the results.

## 2. Conditioned Nonequilibrium Work Relations and Sagawa–Ueda Fluctuation Theorem

We consider a system in contact with a heat bath of inverse temperate $\beta :=1/(k_{B}T)$ where $k_{B}$ is the Boltzmann constant, and *T* is the temperature of the heat bath. External parameter $\lambda_{t}$ drives the system away from equilibrium during 0≤t≤τ. We assume that the initial probability distribution is equilibrium one at control parameter $\lambda_{0}$. Let Γ be the set of all microscopic trajectories, and $\Gamma_{x_{\tau }}$ be that of paths conditioned on $x_{\tau }$ at time τ. Then, the Jarzynski equality [1] and end-point conditioned version [27,28] of it read as follows:(1)$$\begin{array}{ccc} F_{eq}\left(\lambda_{\tau }\right) & = & F_{eq}\left(\lambda_{0}\right)-\frac{1}{\beta }\ln{\langle e^{-\beta W}\rangle }_{\Gamma }\;\mathrm{and} \end{array}$$
(2)$$\begin{array}{ccc} F(x_{\tau },\tau ) & = & F_{eq}\left(\lambda_{0}\right)-\frac{1}{\beta }\ln{\langle e^{-\beta W}\rangle }_{\Gamma_{x_{\tau }}}, \end{array}$$
respectively, where brackets ${\langle \cdot \rangle }_{\Gamma }$ indicates the average over all trajectories in Γ and ${\langle \cdot \rangle }_{\Gamma_{x_{\tau }}}$ indicates the average over trajectories reaching $x_{\tau }$ at time τ. Here *W* indicates work done on the system through $\lambda_{t}$, $F_{eq}\left(\lambda_{t}\right)$ is equilibrium free energy at control parameter $\lambda_{t}$, and $F(x_{\tau },\tau )$ is local non-equilibrium free energy of $x_{\tau }$ at time τ. Work measurement over a specific ensemble of paths gives us equilibrium free energy as a function of $\lambda_{\tau }$ through Equation (Equation 1) and local non-equilibrium free energy as a micro-state function of $x_{\tau }$ at time τ through Equation (2). The following fluctuation theorem links Equations (Equation 1) and (2):(3)$${\langle e^{-\beta F(x_{\tau },\tau )}\rangle }_{x_{\tau }}=e^{-\beta F_{eq}\left(\lambda_{\tau }\right)},$$
where brackets ${\langle \cdot \rangle }_{x_{\tau }}$ indicates the average over all microstates $x_{\tau }$ at time τ [27,28]. Defining the reverse process by $\lambda_{t}^{\prime }:=\lambda_{\tau -t}$ for 0≤t≤τ, the Crooks fluctuation theorem [2] and end-point conditioned version [27,28] of it read as follows:(4)$$\begin{array}{ccc} \frac{P_{\Gamma }\left(W\right)}{P_{\Gamma }^{\prime }\left(-W\right)} & = & \exp\left(\frac{W-\Delta F_{eq}}{k_{B}T}\right)\;\mathrm{and} \end{array}$$
(5)$$\begin{array}{ccc} \frac{P_{\Gamma_{x_{\tau }}}\left(W\right)}{P_{\Gamma_{x_{\tau }}}^{\prime }\left(-W\right)} & = & \exp\left(\frac{W-\Delta F}{k_{B}T}\right), \end{array}$$
respectively, where $P_{\Gamma }\left(W\right)$ and $P_{\Gamma_{x_{\tau }}}\left(W\right)$ are probability distributions of work *W* normalized over all paths in Γ and $\Gamma_{x_{\tau }}$, respectively. Here $P^{\prime }$ indicates corresponding probabilities for the reverse process. For Equation (Equation 4), the initial probability distribution of the reverse process is an equilibrium one at control parameter $\lambda_{\tau }$. On the other hand, for Equation (5), the initiail probability distribution for the reverse process should be non-equilibrium probability distribution $p(x_{\tau },\tau )$ of the forward process at control parameter $\lambda_{\tau }$. By identifying such *W* that $P_{\Gamma }\left(W\right)=P_{\Gamma }^{\prime }\left(-W\right)$, one obtains $\Delta F_{eq}:=F_{eq}\left(\lambda_{\tau }\right)-F_{eq}\left(\lambda_{0}\right)$, the difference in equilibrium free energy between $\lambda_{0}$ and $\lambda_{\tau }$, through Equation (Equation 4) [9]. Similar identification may provide $\Delta F:=F\left(x_{\tau },\tau \right)-F_{eq}\left(\lambda_{0}\right)$ through Equation (5).

Now we turn to the Sagawa–Ueda fluctuation theorem of information exchange [25]. Specifically, we discuss the generalized version [26] of it. To this end, we consider two subsystems *X* and *Y* in the heat bath of inverse temperature β. During process $\lambda_{t}$, they interact and co-evolve with each other. Then, the fluctuation theorem of information exchange reads as follows:(6)$${\langle e^{-\sigma +\Delta I}\rangle }_{\Gamma }=1,$$
where brackets indicate the ensemble average over all paths of the combined subsystems, and σ is the sum of entropy production of system *X*, system *Y*, and the heat bath, and ΔI is the change in mutual information between *X* and *Y*. We note that in the original version of the Sagawa–Ueda fluctuation theorem, only system *X* is in contact with the heat bath and *Y* does not evolve during the process [25,26]. In this paper, we prove an end-point conditioned version of Equation (Equation 6):(7)$$I_{\tau }\left(x_{\tau },y_{\tau }\right)=-\ln{\langle e^{-(\sigma +I_{0})}\rangle }_{x_{\tau },\;y_{\tau }},$$
where brackets indicate the ensemble average over all paths to $x_{\tau }$ and $y_{\tau }$ at time τ, and $I_{t}$ (0≤t≤τ) is local form of mutual information between microstates of *X* and *Y* at time *t* (see Figure 1b). If there is no initial correlation, i.e., $I_{0}=0$, Equation (Equation 7) clearly indicates that local mutual information $I_{\tau }$ as a function of correlated-microstates $\left(x_{\tau },y_{\tau }\right)$ encodes entropy production σ within the end-point conditioned ensemble of paths. In the same vein, we may interpret initial correlation $I_{0}$ as encoded entropy production for the preparation of the initial condition.

## 3. Results

### 3.1. Theoretical Framework

Let *X* and *Y* be finite classical stochastic systems in the heat bath of inverse temperate β. We allowed external parameter $\lambda_{t}$ drives one or both subsystems away from equilibrium during time 0≤t≤τ [29,30,31]. We assumed that classical stochastic dynamics describes the time evolution of *X* and *Y* during process $\lambda_{t}$ along trajectories $\left\{x_{t}\right\}$ and $\left\{y_{t}\right\}$, respectively, where $x_{t}$ ($y_{t}$) denotes a specific microstate of *X* (*Y*) at time *t* for 0≤t≤τ on each trajectory. Since trajectories fluctuate, we repeated process $\lambda_{t}$ with initial joint probability distribution $P_{0}\left(x,y\right)$ over all microstates (x,y) of systems *X* and *Y*. Then the subsystems may generate a joint probability distribution $P_{t}\left(x,y\right)$ for 0≤t≤τ. Let $P_{t}\left(x\right):=\int P_{t}\left(x,y\right)\;dy$ and $P_{t}\left(y\right):=\int P_{t}\left(x,y\right)\;dx$ be the corresponding marginal probability distributions. We assumed
(8)$$P_{0}\left(x,y\right)\neq 0\mathrm{for}\;\mathrm{all}\left(x,y\right),$$
so that we have $P_{t}\left(x,y\right)\neq 0$, $P_{t}\left(x\right)\neq 0$, and $P_{t}\left(y\right)\neq 0$ for all *x* and *y* during 0≤t≤τ. Now we consider entropy production σ of system *X* along $\left\{x_{t}\right\}$, system *Y* along $\left\{y_{t}\right\}$, and heat bath $Q_{b}$ during process $\lambda_{t}$ for 0≤t≤τ as follows
(9)$$\sigma :=\Delta s+\beta Q_{b},$$
where
(10)$$\begin{array}{c} \begin{array}{cc} \Delta s & :=\Delta s_{x}+\Delta s_{y},\\ \Delta s_{x} & :=-\ln P_{\tau }\left(x_{\tau }\right)+\ln P_{0}\left(x_{0}\right),\\ \Delta s_{y} & :=-\ln P_{\tau }\left(y_{\tau }\right)+\ln P_{0}\left(y_{0}\right). \end{array} \end{array}$$

We remark that Equation (Equation 10) is different from the change of stochastic entropy of combined super-system composed of *X* and *Y*, which reads $\ln P_{0}\left(x_{0},y_{0}\right)-\ln P_{\tau }\left(x_{\tau },y_{\tau }\right)$ that reduces to Equation (Equation 10) if processes $\left\{x_{t}\right\}$ and $\left\{y_{t}\right\}$ are independent. The discrepancy leaves room for correlation Equation (Equation 11) below [25]. Here the stochastic entropy $s\left[P_{t}\left(\circ \right)\right]:=-\ln P_{t}\left(\circ \right)$ of microstate ○ at time *t* is uncertainty of ○ at time *t*: the more uncertain that microstate ○ occurs, the greater the stochastic entropy of ○ is. We also note that in [25], system *X* was in contact with the heat reservoir, but system *Y* was not. Nor did system *Y* evolve. Thus their entropy production reads $\sigma_{\mathrm{su}}:=\Delta s_{x}+\beta Q_{b}$.

Now we assume, during process $\lambda_{t}$, that system *X* exchanged information with system *Y*. By this, we mean that trajectory $\left\{x_{t}\right\}$ of system *X* evolved depending on the trajectory $\left\{y_{t}\right\}$ of system *Y* (see Figure 1b). Then, the local form of mutual information $I_{t}$ at time *t* between $x_{t}$ and $y_{t}$ is the reduction of uncertainty of $x_{t}$ due to given $y_{t}$ [25]:(11)$$\begin{array}{c} \begin{array}{cc} I_{t}\left(x_{t},y_{t}\right) & :=s\left[P_{t}\left(x_{t}\right)\right]-s\left[P_{t}\left(x_{t}|y_{t}\right)\right]\\ & \;\;=\ln\frac{P_{t}\left(x_{t},y_{t}\right)}{P_{t}\left(x_{t}\right)P_{t}\left(y_{t}\right)}, \end{array} \end{array}$$
where $P_{t}\left(x_{t}|y_{t}\right)$ is the conditional probability distribution of $x_{t}$ given $y_{t}$. The more information was being shared between $x_{t}$ and $y_{t}$ for their occurrence, the larger the value of $I_{t}\left(x_{t},y_{t}\right)$ was. We note that if $x_{t}$ and $y_{t}$ were independent at time *t*, $I_{t}\left(x_{t},y_{t}\right)$ became zero. The average of $I_{t}\left(x_{t},y_{t}\right)$ with respect to $P_{t}\left(x_{t},y_{t}\right)$ over all microstates is the mutual information between the two subsystems, which was greater than or equal to zero [32].

### 3.2. Proof of Fluctuation Theorem of Information Exchange Conditioned on a Correlated-Microstates

Now we are ready to prove the fluctuation theorem of information exchange conditioned on a correlated-microstates. We define reverse process $\lambda_{t}^{\prime }:=\lambda_{\tau -t}$ for 0≤t≤τ, where the external parameter is time-reversed [33,34]. The initial probability distribution $P_{0}^{\prime }\left(x,y\right)$ for the reverse process should be the final probability distribution for the forward process $P_{\tau }\left(x,y\right)$ so that we have
(12)$$\begin{array}{c} \begin{array}{c} P_{0}^{\prime }\left(x\right)=\int P_{0}^{\prime }\left(x,y\right)\;dy=\int P_{\tau }\left(x,y\right)\;dy=P_{\tau }\left(x\right),\\ P_{0}^{\prime }\left(y\right)=\int P_{0}^{\prime }\left(x,y\right)\;dx=\int P_{\tau }\left(x,y\right)\;dx=P_{\tau }\left(y\right). \end{array} \end{array}$$

Then, by Equation (Equation 8), we have $P_{t}^{\prime }\left(x,y\right)\neq 0$, $P_{t}^{\prime }\left(x\right)\neq 0$, and $P_{t}^{\prime }\left(y\right)\neq 0$ for all *x* and *y* during 0≤t≤τ. For each trajectories $\left\{x_{t}\right\}$ and $\left\{y_{t}\right\}$ for 0≤t≤τ, we define the time-reversed conjugate as follows:(13)$$\begin{array}{c} \begin{array}{c} \left\{x_{t}^{\prime }\right\}:=\left\{x_{\tau -t}^{*}\right\},\\ \left\{y_{t}^{\prime }\right\}:=\left\{y_{\tau -t}^{*}\right\}, \end{array} \end{array}$$
where ∗ denotes momentum reversal. Let Γ be the set of all trajectories $\left\{x_{t}\right\}$ and $\left\{y_{t}\right\}$, and $\Gamma_{x_{\tau },y_{\tau }}$ be that of trajectories conditioned on correlated-microstates $\left(x_{\tau },y_{\tau }\right)$ at time τ. Due to time-reversal symmetry of the underlying microscopic dynamics, the set $\Gamma^{\prime }$ of all time-reversed trajectories was identical to Γ, and the set $\Gamma_{x_{0}^{\prime },y_{0}^{\prime }}^{\prime }$ of time-reversed trajectories conditioned on $x_{0}^{\prime }$ and $y_{0}^{\prime }$ was identical to $\Gamma_{x_{\tau },y_{\tau }}$. Thus we may use the same notation for both forward and backward pairs. We note that the path probabilities $P_{\Gamma }$ and $P_{\Gamma_{x_{\tau },y_{\tau }}}$ were normalized over all paths in Γ and $\Gamma_{x_{\tau },y_{\tau }}$, respectively (see Figure 1a). With this notation, the microscopic reversibility condition that enables us to connect the probability of forward and reverse paths to dissipated heat reads as follows [2,35,36,37]:(14)$$\frac{P_{\Gamma }\left(\left\{x_{t}\right\},\left\{y_{t}\right\}|x_{0},y_{0}\right)}{P_{\Gamma }^{\prime }\left(\left\{x_{t}^{\prime }\right\},\left\{y_{t}^{\prime }\right\}|x_{0}^{\prime },y_{0}^{\prime }\right)}=e^{\beta Q_{b}},$$
where $P_{\Gamma }\left(\left\{x_{t}\right\},\left\{y_{t}\right\}|x_{0},y_{0}\right)$ is the conditional joint probability distribution of paths $\left\{x_{t}\right\}$ and $\left\{y_{t}\right\}$ conditioned on initial microstates $x_{0}$ and $y_{0}$, and $P_{\Gamma }^{\prime }\left(\left\{x_{t}^{\prime }\right\},\left\{y_{t}^{\prime }\right\}|x_{0}^{\prime },y_{0}^{\prime }\right)$ is that for the reverse process. Now we restrict our attention to those paths that are in $\Gamma_{x_{\tau },y_{\tau }}$, and divide both numerator and denominator of the left-hand side of Equation (Equation 14) by $P_{\tau }\left(x_{\tau },y_{\tau }\right)$. Since $P_{\tau }\left(x_{\tau },y_{\tau }\right)$ is identical to $P_{0}^{\prime }\left(x_{0}^{\prime },y_{0}^{\prime }\right)$, Equation (Equation 14) becomes as follows:(15)$$\begin{array}{ccc} \frac{P_{\Gamma_{x_{\tau },y_{\tau }}}\left(\left\{x_{t}\right\},\left\{y_{t}\right\}|x_{0},y_{0}\right)}{P_{\Gamma_{x_{\tau },y_{\tau }}}^{\prime }\left(\left\{x_{t}^{\prime }\right\},\left\{y_{t}^{\prime }\right\}|x_{0}^{\prime },y_{0}^{\prime }\right)} & = & e^{\beta Q_{b}}, \end{array}$$
since the probability of paths is now normalized over $\Gamma_{x_{\tau },y_{\tau }}$. Then we have the following:(16)$$\begin{array}{ccc} \frac{P_{\Gamma_{x_{\tau },y_{\tau }}}^{\prime }\left(\left\{x_{t}^{\prime }\right\},\left\{y_{t}^{\prime }\right\}\right)}{P_{\Gamma_{x_{\tau },y_{\tau }}}\left(\left\{x_{t}\right\},\left\{y_{t}\right\}\right)} & = & \frac{P_{\Gamma_{x_{\tau },y_{\tau }}}^{\prime }\left(\left\{x_{t}^{\prime }\right\},\left\{y_{t}^{\prime }\right\}|x_{0}^{\prime },y_{0}^{\prime }\right)}{P_{\Gamma_{x_{\tau },y_{\tau }}}\left(\left\{x_{t}\right\},\left\{y_{t}\right\}|x_{0},y_{0}\right)}\cdot \frac{P_{0}^{\prime }\left(x_{0}^{\prime },y_{0}^{\prime }\right)}{P_{0}\left(x_{0},y_{0}\right)} \end{array}$$
(17)$$\begin{array}{cc} = & \frac{P_{\Gamma_{x_{\tau },y_{\tau }}}^{\prime }\left(\left\{x_{t}^{\prime }\right\},\left\{y_{t}^{\prime }\right\}|x_{0}^{\prime },y_{0}^{\prime }\right)}{P_{\Gamma_{x_{\tau },y_{\tau }}}\left(\left\{x_{t}\right\},\left\{y_{t}\right\}|x_{0},y_{0}\right)}\cdot \frac{P_{0}^{\prime }\left(x_{0}^{\prime },y_{0}^{\prime }\right)}{P_{0}^{\prime }\left(x_{0}^{\prime }\right)p_{0}^{\prime }\left(y_{0}^{\prime }\right)}\cdot \frac{P_{0}\left(x_{0}\right)P_{0}\left(y_{0}\right)}{P_{0}\left(x_{0},y_{0}\right)}\\ & \;\;\;\times \frac{P_{0}^{\prime }\left(x_{0}^{\prime }\right)}{P_{0}\left(x_{0}\right)}\cdot \frac{P_{0}^{\prime }\left(y_{0}^{\prime }\right)}{P_{0}\left(y_{0}\right)} \end{array}$$
(18)$$\begin{array}{cc} = & \exp\{-\beta Q_{b}+I_{\tau }\left(x_{\tau },y_{\tau }\right)-I_{0}\left(x_{0},y_{0}\right)-\Delta s_{x}-\Delta s_{y}\} \end{array}$$
(19)$$\begin{array}{cc} = & \exp\{-\sigma +I_{\tau }\left(x_{\tau },y_{\tau }\right)-I_{0}\left(x_{0},y_{0}\right)\}. \end{array}$$

To obtain Equation (17) from Equation (Equation 16), we multiply Equation (Equation 16) by $\frac{P_{0}^{\prime }\left(x_{0}^{\prime }\right)P_{0}^{\prime }\left(y_{0}^{\prime }\right)}{P_{0}^{\prime }\left(x_{0}^{\prime }\right)P_{0}^{\prime }\left(y_{0}^{\prime }\right)}$ and $\frac{P_{0}\left(x_{0}\right)P_{0}\left(y_{0}\right)}{P_{0}\left(x_{0}\right)P_{0}\left(y_{0}\right)}$, which are 1. We obtain Equation (18) by applying Equations (Equation 10)–(Equation 12) and (Equation 15) to Equation (17). Finally, we use Equation (Equation 9) to obtain Equation (19) from Equation (18). Now we multiply both sides of Equation (19) by $e^{-I_{\tau }\left(x_{\tau },y_{\tau }\right)}$ and $P_{\Gamma_{x_{\tau },y_{\tau }}}\left(\left\{x_{t}\right\},\left\{y_{t}\right\}\right)$, and take integral over all paths in $\Gamma_{x_{\tau },y_{\tau }}$ to obtain the fluctuation theorem of information exchange conditioned on a correlated-microstates:(20)$$\begin{array}{c} \begin{array}{cc} {\langle e^{-(\sigma +I_{0})}\rangle }_{x_{\tau },y_{\tau }} & :=\int_{\left\{x_{t}\right\},\left\{y_{t}\right\}\in \Gamma_{\left\{x_{\tau }\right\},\left\{y_{\tau }\right\}}}e^{-(\sigma +I_{0})}P_{\Gamma_{x_{\tau },y_{\tau }}}\left(\left\{x_{t}\right\},\left\{y_{t}\right\}\right)\;d\left\{x_{t}\right\}d\left\{y_{t}\right\}\\ & =\int_{\left\{x_{t}\right\},\left\{y_{t}\right\}\in \Gamma_{\left\{x_{\tau }\right\},\left\{y_{\tau }\right\}}}e^{-I_{\tau }\left(x_{\tau },y_{\tau }\right)}P_{\Gamma_{x_{\tau },y_{\tau }}}^{\prime }\left(\left\{x_{t}^{\prime }\right\},\left\{y_{t}^{\prime }\right\}\right)\;d\left\{x_{t}^{\prime }\right\}d\left\{y_{t}^{\prime }\right\}\\ & =e^{-I_{\tau }\left(x_{\tau },y_{\tau }\right)}\int_{\left\{x_{t}\right\},\left\{y_{t}\right\}\in \Gamma_{\left\{x_{\tau }\right\},\left\{y_{\tau }\right\}}}P_{\Gamma_{x_{\tau },y_{\tau }}}^{\prime }\left(\left\{x_{t}^{\prime }\right\},\left\{y_{t}^{\prime }\right\}\right)\;d\left\{x_{t}^{\prime }\right\}d\left\{y_{t}^{\prime }\right\}\\ & =e^{-I_{\tau }\left(x_{\tau },y_{\tau }\right)}. \end{array} \end{array}$$

Here we use the fact that $e^{-I_{\tau }\left(x_{\tau },y_{\tau }\right)}$ is constant for all paths in $\Gamma_{x_{\tau },y_{\tau }}$, probability distribution $P_{\Gamma_{x_{\tau },y_{\tau }}}^{\prime }$ is normalized over all paths in $\Gamma_{x_{\tau },y_{\tau }}$, and $d\left\{x_{t}\right\}=d\left\{x_{t}^{\prime }\right\}$ and $d\left\{y_{t}\right\}=d\left\{y_{t}^{\prime }\right\}$ due to the time-reversal symmetry [38]. Equation (Equation 20) clearly shows that just as local free energy encodes work [27], and local entropy encodes heat [28], the local form of mutual information between correlated-microstates $\left(x_{\tau },y_{\tau }\right)$ encodes entropy production, within the ensemble of paths that reach each microstate. The following corollary provides more information on entropy production in terms of energetic costs.

### 3.3. Corollary

To discuss entropy production in terms of energetic costs, we define local free energy $F_{x}$ of $x_{t}$ and $F_{y}$ of $y_{t}$ at control parameter $\lambda_{t}$ as follows:(21)$$\begin{array}{c} \begin{array}{cc} F_{x}\left(x_{t},t\right) & :=E_{x}\left(x_{t},t\right)-k_{B}Ts\left[P_{t}\left(x_{t}\right)\right]\\ F_{y}\left(y_{t},t\right) & :=E_{y}\left(y_{t},t\right)-k_{B}Ts\left[P_{t}\left(y_{t}\right)\right], \end{array} \end{array}$$
where *T* is the temperature of the heat bath, $k_{B}$ is the Boltzmann constant, $E_{x}$ and $E_{y}$ are internal energy of systems *X* and *Y*, respectively, and $s\left[P_{t}\left(\circ \right)\right]:=-\ln P_{t}\left(\circ \right)$ is stochastic entropy [2,3]. Work done on either one or both systems through process $\lambda_{t}$ is expressed by the first law of thermodynamics as follows:(22)$$W:=\Delta E+Q_{b},$$
where ΔE is the change in internal energy of the total system composed of *X* and *Y*. If we assume that systems *X* and *Y* are weakly coupled, in that interaction energy between *X* and *Y* is negligible compared to the internal energy of *X* and *Y*, we may have
(23)$$\Delta E:=\Delta E_{x}+\Delta E_{y},$$
where $\Delta E_{x}:=E_{x}\left(x_{\tau },\tau \right)-E_{x}\left(x_{0},0\right)$ and $\Delta E_{y}:=E_{y}\left(y_{\tau },\tau \right)-E_{y}\left(y_{0},0\right)$ [39]. We rewrite Equation (18) by adding and subtracting the change of internal energy $\Delta E_{x}$ of *X* and $\Delta E_{y}$ of *Y* as follows:(24)$$\begin{array}{ccc} \frac{P_{\Gamma_{x_{\tau },y_{\tau }}}^{\prime }\left(\left\{x_{t}^{\prime }\right\},\left\{y_{t}^{\prime }\right\}\right)}{P_{\Gamma_{x_{\tau },y_{\tau }}}\left(\left\{x_{t}\right\},\left\{y_{t}\right\}\right)} & = & \exp\{-\beta \left(Q_{b}+\Delta E_{x}+\Delta E_{y}\right)+\beta \Delta E_{x}-\Delta s_{x}+\beta \Delta E_{y}-\Delta s_{y}\}\\ & & \;\;\times \exp\{I_{\tau }\left(x_{\tau },y_{\tau }\right)-I_{0}\left(x_{0},y_{0}\right)\} \end{array}$$
(25)$$\begin{array}{cc} = & \exp\{-\beta \left(W-\Delta F_{x}-\Delta F_{y}\right)+I_{\tau }\left(x_{\tau },y_{\tau }\right)-I_{0}\left(x_{0},y_{0}\right)\}, \end{array}$$
where we have applied Equations (Equation 21)–(Equation 23) consecutively to Equation (Equation 24) to obtain Equation (25). Here $\Delta F_{x}:=F_{x}\left(x_{\tau },\tau \right)-F_{x}\left(x_{0},0\right)$ and $\Delta F_{y}:=F_{y}\left(y_{\tau },\tau \right)-F_{y}\left(y_{0},0\right)$. Now we multiply both sides of Equation (25) by $e^{-I_{\tau }\left(x_{\tau },y_{\tau }\right)}$ and $P_{\Gamma_{x_{\tau },y_{\tau }}}\left(\left\{x_{t}\right\},\left\{y_{t}\right\}\right)$, and take integral over all paths in $\Gamma_{x_{\tau },y_{\tau }}$ to obtain the following:(26)$$\begin{array}{c} \begin{array}{cc} {\langle e^{-\beta \left(W-\Delta F_{x}-\Delta F_{y}\right)-I_{0}}\rangle }_{x_{\tau },y_{\tau }} & :=\int_{\left\{x_{t}\right\},\left\{y_{t}\right\}\in \Gamma_{\left\{x_{\tau }\right\},\left\{y_{\tau }\right\}}}e^{-\beta \left(W-\Delta F_{x}-\Delta F_{y}\right)-I_{0}}P_{\Gamma_{x_{\tau },y_{\tau }}}\left(\left\{x_{t}\right\},\left\{y_{t}\right\}\right)\;d\left\{x_{t}\right\}d\left\{y_{t}\right\}\\ & =\int_{\left\{x_{t}\right\},\left\{y_{t}\right\}\in \Gamma_{\left\{x_{\tau }\right\},\left\{y_{\tau }\right\}}}e^{-I_{\tau }\left(x_{\tau },y_{\tau }\right)}P_{\Gamma_{x_{\tau },y_{\tau }}}^{\prime }\left(\left\{x_{t}^{\prime }\right\},\left\{y_{t}^{\prime }\right\}\right)\;d\left\{x_{t}^{\prime }\right\}d\left\{y_{t}^{\prime }\right\}\\ & =e^{-I_{\tau }\left(x_{\tau },y_{\tau }\right)}, \end{array} \end{array}$$
which generalizes known relations in the literature [24,39,40,41,42,43]. We note that Equation (Equation 26) holds under the weak-coupling assumption between systems *X* and *Y* during process $\lambda_{t}$, and $\Delta F_{x}+\Delta F_{y}$ in Equation (Equation 26) is the difference in non-equilibrium free energy, which is different from the change in equilibrium free energy that appears in similar relations in the literature [24,40,41,42,43]. If there is no initial correlation, i.e., $I_{0}=0$, Equation (Equation 26) indicates that local mutual information $I_{\tau }$ as a state function of correlated-microstates $\left(x_{\tau },y_{\tau }\right)$ encodes entropy production, $\beta (W-\Delta F_{x}-\Delta F_{y})$, within the ensemble of paths in $\Gamma_{x_{\tau },y_{\tau }}$. In the same vein, we may interpret initial correlation $I_{0}$ as encoded entropy-production for the preparation of the initial condition.

In [25], they showed that the entropy of *X* can be decreased without any heat flow due to the negative mutual information change under the assumption that one of the two systems does not evolve in time. Equation (Equation 20) implies that the negative mutual information change can decrease the entropy of *X* and that of *Y* simultaneously without any heat flow by the following:(27)$${\langle \Delta s_{x}+\Delta s_{y}\rangle }_{x_{\tau },y_{\tau }}\geq \Delta I_{\tau }\left(x_{\tau },y_{\tau }\right),$$
provided ${\langle Q_{b}\rangle }_{x_{\tau },y_{\tau }}=0$. Here $\Delta I_{\tau }\left(x_{\tau },y_{\tau }\right):=I_{\tau }\left(x_{\tau },y_{\tau }\right)-{\langle I_{0}\left(x_{0},y_{0}\right)\rangle }_{x_{0},y_{0}}$. In terms of energetics, Equation (Equation 26) implies that the negative mutual information change can increase the free energy of *X* and that of *Y* simultaneously without any external-supply of energy by the following:(28)$$-\Delta I_{\tau }\left(x_{\tau },y_{\tau }\right)\geq \beta {\langle \Delta F_{x}+\Delta F_{y}\rangle }_{x_{\tau },y_{\tau }}$$
provided ${\langle W\rangle }_{x_{\tau },y_{\tau }}=0$.

## 4. Examples

### 4.1. A Simple One

Let *X* and *Y* be two systems that weakly interact with each other, and be in contact with the heat bath of inverse temperature β. We may think of *X* and *Y*, for example, as bio-molecules that interact with each other or *X* as a device which measures the state of other system and *Y* be a measured system. We consider a dynamic coupling process as follows: Initially, *X* and *Y* are separately in equilibrium such that the initial correlation $I_{0}\left(x_{0},y_{0}\right)$ is zero for all $x_{0}$ and $y_{0}$. At time t=0, system *X* starts (weak) interaction with system *Y* until time t=τ. During the coupling process, external parameter $\lambda_{t}$ for 0≤t≤τ may exchange work with either one or both systems (see Figure 1b). Since each process fluctuates, we repeat the process many times to obtain probability distribution $P_{t}\left(x,y\right)$ for 0≤t≤τ. We allow both systems co-evolve interactively and thus $I_{t}\left(x_{t},y_{t}\right)$ may vary not necessarily monotonically. Let us assume that the final probability distribution $P_{\tau }\left(x_{\tau },y_{\tau }\right)$ is as shown in Table 1.

Then, a few representative mutual information read as follows:(29)$$\begin{array}{c} \begin{array}{cc} I_{\tau }\left(x_{\tau }=0,y_{\tau }=0\right) & =\ln\frac{1/6}{\left(1/3)\cdot (1/3\right)}=\ln\left(3/2\right),\\ I_{\tau }\left(x_{\tau }=0,y_{\tau }=1\right) & =\ln\frac{1/9}{\left(1/3)\cdot (1/3\right)}=0,\\ I_{\tau }\left(x_{\tau }=0,y_{\tau }=2\right) & =\ln\frac{1/18}{\left(1/3)\cdot (1/3\right)}=\ln\left(1/2\right). \end{array} \end{array}$$

By Jensen’s inequality [32], Equation (Equation 20) implies
(30)$${\langle \sigma \rangle }_{x_{\tau },y_{\tau }}\geq I_{\tau }\left(x_{\tau },y_{\tau }\right).$$

Thus coupling $x_{\tau }=0,y_{\tau }=0$ accompanies on average entropy production of at least ln(3/2) which is greater than 0. Coupling $x_{\tau }=0,y_{\tau }=1$ may not produce entropy on average. Coupling $x_{\tau }=0,y_{\tau }=2$ on average may produce negative entropy by ln(1/2)=−ln2. Three individual inequalities provide more detailed information than that from ${\langle \sigma \rangle }_{\Gamma }\geq {\langle I_{\tau }\left(x_{\tau },y_{\tau }\right)\rangle }_{\Gamma }\approx 0.0872$ currently available from [25,26].

### 4.2. A “Tape-Driven” Biochemical Machine

In [44], McGrath et al. proposed a physically realizable device that exploits or creates mutual information, depending on system parameters. The system is composed of an enzyme *E* in a chemical bath, interacting with a tape that is decorated with a set of pairs of molecules (see Figure 2a). A pair is composed of substrate molecule *X* (or phosphorylated $X^{*}$) and activator *Y* of the enzyme (or $\bar{Y}$ which denotes the absence of *Y*). The binding of molecule *Y* to *E* converts the enzyme into active mode $E^{†}$, which catalyzes phosphate exchange between ATP and *X*:
(31)$$X+\mathrm{ATP}+E^{†}⇌E^{†}\text{-}X\text{-}\mathrm{ADP}\text{-}P_{i}⇌E^{†}+X^{*}+\mathrm{ADP}.$$

The tape is prepared in a correlated manner through a single parameter Ψ:
(32)$$\begin{array}{c} \begin{array}{cc} p_{0}\left(\bar{Y}|X^{*}\right) & =p_{0}\left(Y|X\right)=\Psi ,\\ p_{0}\left(Y|X^{*}\right) & =p_{0}\left(\bar{Y}|X\right)=1-\Psi . \end{array} \end{array}$$

If Ψ<0.5, a pair of *Y* and $X^{*}$ is abundant so that the interaction of enzyme *E* with molecule *Y* activates the enzyme, causing the catalytic reaction of Equation (Equation 31) from the right to the left, resulting in the production of ATP from ADP. If the bath were prepared such that [ATP]>[ADP], the reaction corresponds to work on the chemical bath against the concentration gradient. Note that this interaction causes the conversion of $X^{*}$ to *X*, which reduces the initial correlation between $X^{*}$ and *Y*, resulting in the conversion of mutual information into work. If *E* interacts with a pair of $\bar{Y}$ and *X* which is also abundant for Ψ<0.5, the enzyme becomes inactive due to the absence of *Y*, preventing the reaction Equation (Equation 31) from the left to the right, which plays as a ratchet that blocks the conversion of *X* and ATP to $X^{*}$ and ADP, which might happen otherwise due to the the concentration gradient of the bath.

On the other hand, if Ψ>0.5, a pair of *Y* and *X* is abundant which allows the enzyme to convert *X* into $X^{*}$ using the pressure of the chemical bath, creating the correlation between *Y* and $X^{*}$. If *E* interacts with a pair of $\bar{Y}$ and $X^{*}$ which is also abundant for Ψ>0.5, the enzyme is again inactive, preventing the de-phosphorylation of $X^{*}$, keeping the created correlation. In this regime, the net effect is the conversion of work (due to the chemical gradient of the bath) to mutual information. The concentration of ATP and ADP in the chemical bath is adjusted via α∈(−1,1) such that
(33)[ATP]=1+αand[ADP]=1−α
relative to a reference concentration $C_{0}$. For the analysis of various regimes of different parameters, we refer the reader to [44].

In this example, we concentrate on the case with α=0.99 and Ψ=0.69, where Ref. [44] pays a special attention. They analyzed the dynamics of mutual information ${\langle I_{t}\rangle }_{\Gamma }$ during $10^{-2}\leq t\leq 10^{2}$. Due to the high initial correlation, the enzyme converts the mutual information between $X^{*}$ and *Y* into work against the pressure of the chemical bath with [ATP]>[ADP]. As the reactions proceed, correlation ${\langle I_{t}\rangle }_{\Gamma }$ drops until the minimum reaches, which is zero. Then, eventually the reaction is inverted, and the bath begins with working to create mutual information between $X^{*}$ and *Y* as shown in Figure 2b.

We split the ensemble $\Gamma^{t}$ of paths into $\Gamma_{X,Y}^{t}$ composed of trajectories reaching (X,Y) at each *t* and $\Gamma_{X^{*},Y}^{t}$ composed of those reaching $\left(X^{*},Y\right)$ at time *t*. Then, we calculate $I_{t}\left(X,Y\right)$ and $I_{t}\left(X^{*},Y\right)$ using the analytic form of probability distributions that they derived. Figure 2c,d show $I_{t}\left(X,Y\right)$ and $I_{t}\left(X^{*},Y\right)$, respectively, as a function of time *t*. During the whole process, mutual information $I_{t}\left(X,Y\right)$ monotonically decreases. For $10^{-2}\leq t\leq 10^{1/3}$, it keeps positive, and after that, it becomes negative which is possible for local mutual information. Trajectories in $\Gamma_{X,Y}$ harness mutual information between $X^{*}$ and *Y*, converting $X^{*}$ to *X* and ADP to ATP against the chemical bath. Contrary to this, $I_{t}\left(X^{*},Y\right)$ increases monotonically. It becomes positive after $t>10^{1/3}$, indicating that the members in $\Gamma_{X^{*},Y}^{t}$ create mutual information between $X^{*}$ and *Y* by converting *X* to $X^{*}$ using the excess of ATP in the chemical bath. The effect accumulates, and the negative values of $I_{t}\left(X^{*},Y\right)$ turn to the positive after $t>10^{1/3}$.

## 5. Conclusions

We have proved the fluctuation theorem of information exchange conditioned on correlated-microstates, Equation (Equation 20), and its corollary, Equation (Equation 26). Those theorems make it clear that local mutual information encodes as a state function of correlated-states entropy production within an ensemble of paths that reach the correlated-states. Equation (Equation 20) also reproduces lower bound of entropy production, Equation (Equation 30), within a subset of path-ensembles, which provides more detailed information than the fluctuation theorem involved in the ensemble of all paths. Equation (Equation 26) enables us to know the exact relationship between work, non-equilibrium free energy, and mutual information. This end-point conditioned version of the theorem also provides more detailed information on the energetics for coupling than current approaches in the literature. This robust framework may be useful to analyze thermodynamics of dynamic molecular information processes [44,45,46] and to analyze dynamic allosteric transitions [47,48].

## Figures and Tables

**Figure 1 entropy-21-00477-f001:**
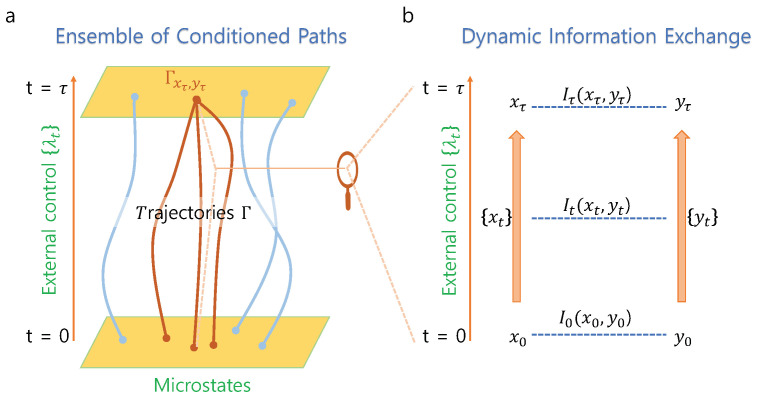
Ensemble of conditioned paths and dynamic information exchange: (**a**) Γ and $\Gamma_{x_{\tau },y_{\tau }}$ denote respectively the set of all trajectories during process $\lambda_{t}$ for 0≤t≤τ and that of paths that reach $\left(x_{\tau },y_{\tau }\right)$ at time τ. Red curves schematically represent some members of $\Gamma_{x_{\tau },y_{\tau }}$. (**b**) We magnified a single trajectory in the left panel to represent a detailed view of dynamic coupling of $\left(x_{\tau },y_{\tau }\right)$ during process $\lambda_{t}$. The point-wise mutual information $I_{t}\left(x_{t},y_{t}\right)$ may vary not necessarily monotonically.

**Figure 2 entropy-21-00477-f002:**
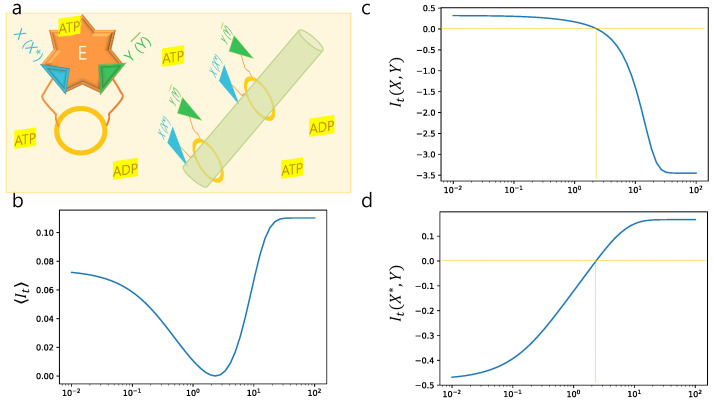
Analysis of a “tape-driven” biochemical machine: (**a**) a schematic illustration of enzyme *E*, pairs of $X(X^{*})$ and $Y(\bar{Y})$ in the chemical bath including ATP and ADP. (**b**) The graph of ${\langle I_{t}\rangle }_{\Gamma }$ as a function of time *t*, which shows the non-monotonicity of ${\langle I_{t}\rangle }_{\Gamma }$. (**c**) The graph of $I_{t}\left(X,Y\right)$ which decreases monotonically and composed of trajectories that harness mutual information to work against the chemical bath. (**d**) The graph of $I_{t}\left(X^{*},Y\right)$ that increases monotonically and composed of paths that create mutual information between $X^{*}$ and *Y*.

**Table 1 entropy-21-00477-t001:** The joint probability distribution of *x* and *y* at final time τ: Here we assume that both systems *X* and *Y* have three states, 0, 1, and 2.

X\Y	0	1	2
**0**	1/6	1/9	1/18
**1**	1/18	1/6	1/9
**2**	1/9	1/18	1/6

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
