# Peer review of "Fluctuation Theorem of Information Exchange within an Ensemble of Paths Conditioned on Correlated-Microstates"

_entropy, 2019, doi:10.3390/e21050477_

Round 1

Reviewer 1 Report

In this article, authors show that mutual information encodes the amount of entropy production of the ensemble paths reaching the coupled states.  Each single microstate is considered as an ensemble of trajectories in local version of fluctuation theorem of information exchange.  Authors start by briefly discussing Crooks and Jarzynski fluctuation theorem, after which they introduce the Ensemble of Conditioned Paths and Dynamic Information Exchange.  In the Corollary, they discuss entropy production in terms of energetic costs for weakly coupled paths followed by an example.

This is a short, but well written article, especially if one considers that the Overview of Fluctuation Theorems is well known fact.  Nevertheless, the clear idea, language, and the useful illustration makes this article a significant contribution.  I recommend publishing this article in Entropy in its present form.

Author Response

Response to Reviewer 1

Comments

First of all, we would like to thank the referee for the very positive responses for the publication of the paper to entropy.

Point 1: In this article, authors show that mutual information encodes the amount of entropy production of the ensemble paths reaching the coupled states.  Each single microstate is considered as an ensemble of trajectories in local version of fluctuation theorem of information exchange.  Authors start by briefly discussing Crooks and Jarzynski fluctuation theorem, after which they introduce the Ensemble of Conditioned Paths and Dynamic Information Exchange.  In the Corollary, they discuss entropy production in terms of energetic costs for weakly coupled paths followed by an example.

Response 1: We agree on this.

Point 2:This is a short, but well written article, especially if one considers that the Overview of Fluctuation Theorems is well known fact.  Nevertheless, the clear idea, language, and the useful illustration makes this article a significant contribution.  I recommend publishing this article in Entropy in its present form.

Response 2: We thank the referee for the recommendation of this article in Entropy.

Author Response

Response to Reviewer 2 

Comments

First of all, we would like to thank the referee for providing very constructive comments that enable to improve the weak points of the paper such that the manuscript may reach the high quality required for acceptance in Entropy. We have modified the document (written in blue for modifications) following the referee's comments as follows.

Point 1: The first sentence of the Abstract, “Fluctuation theorems are a class of equalities each of which links a thermodynamic path functional such as heat and work to a state function such as entropy and free energy”, is incorrect. There are several fluctuation theorems not containing thermodynamic state functions, as comprehensively reviewed in Ref. [17] of the present manuscript. 

Response 1: We modified the sentence more correctly by referring to Ref. [17] as follows:

Fluctuation theorems are a class of equalities that express universal properties of the probability distribution of a fluctuating path functional such as heat, work or entropy production over an ensemble of trajectories during a non-equilibrium process with a well-defined initial distribution.

Point 2:The notion “true thermodynamic potential” used in the abstract is misleading. There is no clear connection between the present concept of “local mutual information” and “true thermodynamic potentials” used in classical thermodynamics, see e.g. the standard textbook by Callen: Thermodynamics and an Introduction to Thermostatistics (2nd ed., John Wiley & Sons). Beyond the abstract, the term “true thermodynamic potential” is only mentioned once (in the Introduction). I recommend to remove the “true thermodynamic potential” from the abstract, unless the author could establish and explain the clear relation to the standard classical notion of thermodynamic potentials (i.e., to show that authors’ concept is related to the thermodynamic entropy via the Legendre transform; to prove the extremalization principle, which will lead to conditions of equilibrium and to stability criteria, as it is done for each true thermodynamic potential).

Response 2: We thank the referee for this comment that makes the conceptual aspect of the paper stronger. We removed the term “true thermodynamic potential” and modified related sentences appropriately.

Point 3:The term “macrostate”, again well known from equilibrium statistical mechanics, is not used correctly in the manuscript. In equilibrium theory, the macrostate is determined by macroscopic state variables (T, P, V, N, ...). We can not say “the macrostate V ”, but instead “the macrostate of the system determined by values of T,V,N”, which implicitly refers to the equilibrium Boltzmann distri- bution. From this perspective, authors’ frequently used “the system macrostate λt” makes no physical sense. In fact, it seems to me that the author confuses the term “macrostate” with the term “protocol”. The latter is commonly used when referring to externally controlled system parameters. Therefore, I recommend not

to use the term “macrostate” in the present manuscript, otherwise a reader may be confused.

Response 3: We thank the referee for this comment that also makes the conceptual aspect of the paper stronger. We removed the term “macrostate” and modified related sentences appropriately.

Point 4:The title of Sec. 2 should be more specific e.g., “Conditioned nonequilibrium work relations and Ueda-Sagawa FT”. The current “Brief Overview of Fluctuation Theorems” is inappropriate, because several important FTs are not mentioned, cf. Ref. [17].

Response 4: We modified the title of Sec.2 following the comment.

Point 5:The term “conditioned at” frequently used in the manuscript is rather unusual in English. Apparently the author means “conditioned on” (in the sense “conditioning on an event”) or/and “conditioned by”.

Response 5: We fixed the term following the comment.

Point 6:It should be clarified, what is the exact meaning of the frequently used “local”. Is it space-local, or time-local, or is it just related to the instantaneous microstate on a system trajectory?

Response 6: We make it clear that local is related to a specific microstate x at time t considered as an ensemble of paths to it. The state as an ensemble is slightly different from the instantaneous microstate on a system trajectory since, for example, the average velocity of x at time t as an ensemble is well-defined for the former but not for the latter. 

Point 7:The reference list lacks several important related works in stochastic thermody- namics of information. These comprise for instance the following:

- pioneering studies on thermodynamics of feedback-controlled systems:

Phys. Rev. A 39, 5378 (1989), DOI: 10.1103/PhysRevA.39.5378

Phys. Rev. E 79, 041118 (2009), DOI: 10.1103/PhysRevE.79.041118

- important discoveries after the pioneering work of Sagawa and Ueda such as the unifying formulation of fluctuation theorems for information processing:

Phys. Rev. Lett. 112, 090601 (2014), DOI: 10.1103/PhysRevLett.112.090601

Phys. Rev. E 90, 042150 (2014), DOI: 10.1103/PhysRevE.90.042150 

- description of continuous information flows

Phys. Rev. X 4, 031015 (2014), DOI: 10.1103/PhysRevX.4.031015

Europhys. Lett. 116, 10007 (2016), DOI: 10.1209/0295-5075/116/10007 

- unifying formalism for quantum thermodynamics of information

Phys. Rev. X 7, 021003 (2017), DOI: 10.1103/PhysRevX.7.021003

to name only a few. It is highly desirable to devote more space to discuss a relation between the present work and the current state of the art in the field. Regarding the important role played by fluctuations in microscopic machines, to which the author refers in the Introduction, recent works addressing this issue theoretically and experimentally, e.g., are:

Phys. Rev. Lett. 121, 120601 (2018), DOI: 10.1103/PhysRevLett.121.120601

Phys. Rev. Lett. 121, 230601 (2018), DOI: 10.1103/PhysRevLett.121.230601 and also see the recent review of experiments in stochastic thermodynamics:

Phys. Rev. X 7, 021051 (2017), DOI: 10.1103/PhysRevX.7.021051

Response 7: We supplemented the vital list for thermodynamics of information in the introduction.

Point 8:Decomposition of the “stochastic entropy” given in Eqs. (10) is valid for indepen- dent processes xtand ytonly. The change of this entropy when xtand ytare not independent (which is the case discussed in the manuscript) is given by

∆s = −lnPτ(xτ,yτ)+lnP0(x0,y0),              (R1)

which reduces to Eqs. (10) only if Pτ(xτ,yτ) = Pτ(xτ)Pτ(yτ) and P0(x0,y0) =P0(x0)P0(y0) hold. This fact should be stated clearly when referring to Eqs. (10).

Response 8: We make this fact clear when referring to Eqs. (10).

Point 9:The example presented in Sec. 4 does not illustrate properly the role of conditioning on the final state (which is the principal new idea of the manuscript). The example should clearly demonstrate why the presented FT is useful and how the conditioning affects bounds on work extractable by feedback. That is, it should treat the dynamics of the both processes (as emphasized by the author, the stochastic dynamics of the controller ytis an important element of the present manuscript), and discuss properties of the presented functionals. Instead, in Sec. 4, only the mutual information of two mutually dependent two-state systems with a prescribed distribution at a single time is calculated. (The latter calculation is a standard textbook example.)

Response 9: We supplemented the example by considering a “tape-driven” biochemical machine [McGrath et al. PRL 118, 028101 (2017)], which consumes or establishes mutual information during a process.

Point 10:There are typos in the manuscript. Additional proofreading is necessary.

Response 10: We have made additional proofreading to fix the typos. 

Round 2

Reviewer 2 Report

Dear Editor: 

The author reacted satisfactorily to all my concerns and modified the manuscript accordingly. In my opinion, the revised manuscript can be accepted for publication in Entropy in its current form.

Author Response

We thank the referee for the recommendation of the article for publication in Entropy.